# New Intranuclear Symbiotic Bacteria from Macronucleus of *Paramecium putrinum*—"*Candidatus* Gortzia Yakutica"

**Alexandra Y. Beliavskaia** [1,*] , **Alexander V. Predeus** [2], **Sofya K. Garushyants** [3],
**Maria D. Logacheva** [4], **Jun Gong** [5,6], **Songbao Zou** [5], **Mikhail S. Gelfand** [3,4]
**and Maria S. Rautian** [1,*]

1   Department of Invertebrate Zoology, Saint Petersburg State University, 199034 Saint Petersburg, Russia
2   Bioinformatics Institute, 197342 Saint Petersburg, Russia; predeus@bioinf.me
3   Kharkevitch Institute for Information Transmission Problems, 127051 Moscow, Russia;
    garushyants@gmail.com (S.K.G.); gelfand@iitp.ru (M.S.G.)
4   Skolkovo Institute of Science and Technology, 143026 Moscow, Russia; maria.log@gmail.com
5   Yantai Institute of Coastal Zone Research, Chinese Academy of Sciences, Yantai 264003, China;
    jgong@yic.ac.cn (J.G.); songbaozou@126.com (S.Z.)
6   School of Marine Sciences, Sun Yat-Sen University, Zhuhai 519082, China
*   Correspondence: alex.beliavskaia@gmail.com (A.Y.B.); mrautian@mail.ru (M.S.R.)

**Abstract:**  *Holospora*-like bacteria (HLB) are obligate intracellular *Alphaproteobacteria*, inhabiting nuclei of *Paramecium* and other ciliates such as "*Candidatus* Hafkinia" is in *Frontonia*.  The HLB clade is comprised of four genera, *Holospora*, *Preeria*, "*Candidatus* Gortzia", and "*Candidatus* Hafkinia". These bacteria have a peculiar life cycle with two morphological forms and some degree of specificity to the host species and the type of nucleus they inhabit.  Here we describe a novel species of HLB—"*Candidatus* Gortzia yakutica" sp. nov.—a symbiont from the macronucleus of *Paramecium putrinum*, the first described HLB for this *Paramecium* species.  The new endosymbiont shows morphological similarities with other HLB. The phylogenetic analysis of the SSU rRNA gene places it into the "*Candidatus* Gortzia" clade.

**Keywords:** symbiosis; intranuclear bacteria; *Holospora*; Gortzia; *Paramecium*

## 1. Introduction

*Paramecium* ciliates (Oligohymenophorea, Ciliophora, Alveolata) host diverse intracellular symbionts, among which the best studied are *Holospora*-like bacteria (HLB), obligate intranuclear bacteria of family *Holosporaceae*, order *Holosporales*, class *Alphaproteobacteria* [1–4]. HLB have a set of interesting features, such as a complex life cycle involving two morphological stages, infectious and reproductive, and infectious forms (IFs) which are unusually large for bacterial cells (up to 20 µm long). IFs have hypertrophied periplasm forming about half of the cell, and a recognition tip on the periplasm end [5]; they can survive in ambient conditions for several hours and infect new host cells. The reproductive forms (RFs) are small and able to reproduce by binary fission, and can transform into IFs [1,6]. HLB species can distinguish between two types of host nuclei, macronucleus (Ma) and micronucleus (Mi) [2].

These features were traditionally used to assign bacteria to genus *Holospora* before any molecular information was available.  Thus, until the emergence of sequencing methods, all bacteria with the described morphological and physiological features were considered *Holospora* species and classified by their host specificity, localization in the host cell, size and shape of IFs and RFs, and the ability to trigger

formation of the connecting piece during division of the infected nucleus [1,7,8]. Infectious forms of *H. obtusa*, *H. undulata*, *H. elegans*, "*H. curviuscula*", and "*H. acuminata*" gather near the center of the spindle apparatus of the dividing host nucleus forming the so-called connecting piece, while the reproductive forms mainly appear in the apical parts of the nucleus. Following formation of the connecting piece, IFs escape into the cytoplasm and then into the ambient environment. The second group of the *Holospora* species ("*H. caryophila*", "*H. bacillata*", and "*H. curvata*") consists of HLB, which do not form the connecting piece [6,8].

With the recent advance of sequencing techniques, the phylogeny of genus *Holospora* has been revised [9–13]. One of the *Holospora* species, *H. caryophila*, was recently redescribed as *Preeria caryophila* based on low similarity of the 16S rRNA gene with other species from genus *Holospora* and the ability to infect several host species [10]. Boscaro et al. recently reported a new genus of HLB, "*Ca*. Gortzia", currently comprised of two species, "*Ca*. Gortzia infectiva" [11], and "*Ca. Gortzia shahrazadis*" [12], macronuclear symbionts of *Paramecium jenningsi* and *Paramecium multimicronucleatum*, respectively. These endosymbionts do not induce the formation of the connecting piece in the nucleus during division of *Paramecium*. "*Ca*. Hafkinia simulans" was recently described by Fokin et al. within *Holosporaceae* as a macronuclear symbiont of a ciliate *Frontonia salmastra* showing typical HLB features [14]. The 16S rRNA gene sequences obtained from "*Ca*. Gortzia" and "*Ca*. Hafkinia" differ by approximately 7–10% from the 16S rRNA of *Holospora* species, which is beyond the threshold for the genus level [11,15]. Together with other discriminating features like inducing formation of the connecting piece (now assigned only to genus *Holospora*), and reduced host specificity as shown for *Preeria caryophila*, it supports the separation of HLB group into four genera [10,11].

Here we report a new *Holospora*-like intranuclear bacterium in the macronucleus of ciliate *Paramecium putrinum* originating from Yakutia (Sakha Republic), Russia. Our microscopical observations, phylogenetic analysis based on the 16S rRNA genes, and fluorescence in situ hybridization assays allow suggestion of its inclusion as a novel member of genus "*Ca*. Gortzia". We suggest this bacterium to be classified as a new species "*Ca. Gortzia yakutica*" sp. n.

## 2. Materials and Methods

### 2.1. Sampling and Identification of Paramecium

The ciliate *P. putrinum* YA111-52 was originally isolated from a freshwater pond in Yakutia (62°02′ N 129°44′ E), Sakha Republic, Russia in the summer of 2013. Monoclonal cultures of this species were maintained under standard conditions at the room temperature in lettuce medium inoculated with bacterium *Enterobacter aerogenes* as the food source [16]. The host was identified by cell morphology, the structure of the micronucleus, and a contractile vacuole [17,18]. Live observations and images were made at the St. Petersburg State University Center for Culturing Collection of Microorganisms with a Leica DM2500 microscope equipped with differential interference contrast (DIC).

The syngen of *P. putrinum* YA111-52 was determined by series of crossing with *P. putrinum* test-clones from two syngens (syngen 1: clones ABT1-3, ALT27-6, syngen 2: clones BBR51-12, YA1-8). All cultures were fed the day before the experiment. Approximately 100 cells of the testing clones were mixed with an equal number of test-clones' cells [19,20].

### 2.2. Phenotypic Characterization of the Symbionts

The infectious capability of the new HLB was proved by adding IFs of the bacteria to a non-infected *P. putrinum* culture. Cross-infection experiments were performed with four *P. putrinum* clones from both syngens listed above. *Paramecium* cells containing IFs of the new HLB were concentrated at 4500 g for 10 min and homogenized using 1% solution of detergent Nonidet P-40 (Sigma-Aldrich Cat No. 21-3277 SAJ). A small amount of the homogenate was checked at 200× magnification to verify that all ciliate cells had been broken. Equal amounts of homogenate were mixed with recipient *Paramecium* cultures and incubated at the room temperature. Cells were observed at 24 and 48 h post-infection.

Additional checks of the mixed cultures were performed every two weeks during the following two months [20].

### 2.3. Purification and Sequencing of Symbionts

The cell culture of *P. putrinum* containing IFs of the new HLB was concentrated and homogenized as stated above. The infectious forms of the endosymbiont were isolated from the homogenate by centrifugation in Percoll density gradient (Sigma-Aldrich, St. Louis, MO, USA, Cat No. P1644) as described previously [13]. DNA from the purified IFs was isolated with the DNeasy Blood and Tissue kit (QIAGEN Cat No. 69504) using a modified protocol as described previously [21].

Bacterial universal primers 27F1 (5′-AGAGTTTGATCCTGGCTCAG-3′) and 1492R (5′-GGTTA CCTTGTTACGACTT-3′) were used for the amplification of 16S rDNA [22]. PCR products were gel purified and cloned with the TIAN Quick Midi Purification Kit (Tiangen, Beijing, China) following the manufacturer's recommendations. Purified rDNA inserted in the PTZ57 RT plasmid vector (InsTAclone PCR Clone Kit, Fermentas), the recombinant plasmids were transformed to competent cells Trans5α (TransGen Biotech, Beijing, China). The positive clones were digested with HhaI (Fermentas, Thermo Scientific, Waltham, MA, USA). Clones determined to be unique by the RFLP analysis were sequenced by an automated ABI DNA sequencer (model 373, PE Applied Biosystems) with primers M13. In this study, 50 positive clones were randomly selected and analyzed using RFLP with enzyme HhaI, 26 unique clones were sequenced.

### 2.4. Fluorescence In Situ Hybridization (FISH)

Fluorescence in situ hybridization (FISH) with rRNA-targeted probes was performed to visualize the localization of the endosymbiont. The probe was designed specifically for the new HLB—Gyak567 (5′-AGGTAGCCACCTACACA-3′). The probe was tested against the SILVA r138 database using TestProbe 3.0 [23] allowing 0 mismatches. There was one match found for the sequence GYAK567 in the REFNR sequence collection belonging to uncultured bacterium clone lp146, environmental sample from apple orchard, China (GenBank KC331364). The efficiency of the probe was tested in silico using mathFISH tool (mathfish.cee.wisc.edu), resulting in $G^o_{overall}$ of $-12.2$ kcal/mol, and 0.9954 hybridization efficiency. The designed probe was found to have at least three mismatches with other *Holospora*-like bacteria shown in the supplementary Figure S1.

The probe was labeled with the cyanine 5 (Cy5) fluorescent dye at the 5′ end. We also used the Eub338 probe for *Bacteria* labeled with Fluorescein as a positive control [24]. *P. putrinum* cell culture containing the new HLB was concentrated using centrifugation at 3000 g for 10 min. Cells were fixed in 4% paraformaldehyde in the 1X PBS buffer at 4 °C for 3 h shaken every 30 min, the cells were pelleted by centrifugation, and washed twice with the PBS solution to remove the residual fixative. The hybridization buffer (0.9 M NaCl, 20 mM Tris-HCI pH 7.2, 0.01% SDS) and the probe stock to the final concentration of 5 ng/μL were added. The hybridization was followed by three 20 min post-hybridization washes at 48°C in the washing buffer (0.9 M NaCl, 20 mM Tris-HCI pH 7.2, 0.01% SDS). Cells were embedded on slides in Mowiol 4-88 mounting medium (Sigma, St. Louis, MO, USA), prepared as described in Cold Spring Harbor protocols [25]. All experiments included a negative control without probes to test for autofluorescence. The slides were imaged with Leica TCS SP5 confocal laser scanning microscope in The Chromas Research Facility at Saint Petersburg State University.

### 2.5. Phylogenetic Analysis

Seventy-nine individual sequences of 16S rRNA genes were used for the phylogenetic analysis of 36 *Rickettsiales*, *Holosporales*, and other related bacteria (see Table S1 for the accession numbers). Seventy-two sequences were obtained from GenBank [26] and seven more were extracted from assembled genomes [27] as follows: all GenBank sequences were used as *BLASTN* queries against seven genome assemblies, then the intervals overlapping high-scoring hits (alignment length > 1300)

were extracted as an interval BED file, merged using *bedtools* v2.29.0 [28] (*bedtools merge*), and the corresponding sequence together with 500 bp flanking on each side was extracted using *bedtools getfasta*. Only Genbank sequences longer than 1200 bp were used.

The initial multiple alignment was constructed using *ssu-align* v0.1.1 [29] with the default settings and then filtered using *ssu-mask* v0.1.1, yielding a multiple sequence alignment (MSA) of 1397 bp (see Figure S2 for the structural analysis of the retained sites). The MSA was further analyzed using *BMGE* v1.12 [30], and the alignment was additionally trimmed to account for the shortest sequences; to this end, 154 bp at the 5' end and 164 at the 3' end were removed, resulting in the final multiple alignment of 1079 bp.

Sequence similarity was calculated from the trimmed MSA using a custom Perl script, and visualized using *ggplot2* [31] and *R*.

To select the model that best fits our data, *modeltest-ng* [32] was run on the trimmed MSA with the default parameters. All three criteria used by *modeltest-ng* (BIC, AIC, AICc) have indicated similar models: GTRGAMMAI was found to be the best model using BIC, and GTRGAMMAIX was selected using AIC/AICc. Therefore, the latter model was selected for the following analysis. RAxML v8.2.12 [33] was run on the MSA using "-m GTRGAMMAIX -f a -x 123 -N 1000 -p 456" options, generating 1000 bootstraps. The resulting phylogenetic tree was visualized using Interactive Tree of Life v4 [34]. The 16S rRNA of the 21 sequenced clones of "*Ca.* Gortzia yakutica" strain YA111-52 were deposited in the GenBank database under the accession numbers MT421875.1–MT421895.1. The resulting phylogenetic tree has clonal and outgroup sequences collapsed, while the complete tree is presented as Figure S3. Additionally, we have run the Bayesian inference of phylogeny using *MrBayes* v3.2.7 [35]. The tree topology comparison ("tanglegram") was generated using *Dendroscope* [36], and is available as Figure S4.

Exact commands used in the analysis, the code to reproduce the visualization, and the analysis scripts are available at https://github.com/apredeus/yakutica.

## 3. Results

### 3.1. Bacterial Morphology and Localization

The new HLB was found in macronucleus of *P. putrinum* YA111-52, isolated in the freshwater pond in Yakutia, Russia (Figure 1A). The infection was stable for at least three years under laboratory conditions. A small number of IFs could be found in the cytoplasm of the host cell (Figure 2), suggesting that there might be an intermediate state before the symbiont release into the environment. The endosymbionts were observed in two morphological forms of their life cycle: small (1–2 × 2–4 μm) bacteria undergoing binary fissions (RFs), and long (1–2 × 7–12 μm) IFs. Most observed IFs had straight rod-like shape with tapered ends, and some were slightly curved (Figure 1B). The symbionts were never observed in the micronucleus both in stably infected cultures and during the infection process. We also never observed the formation of the connecting piece during the host cell division (Figure 3), similar to what was previously described for species of the genus "*Ca.* Gortzia".

The endosymbionts are capable of infecting aposymbiotic cells of *P. putrinum*. IFs reach macronucleus and begin to divide in 20–30 h after infection forming chains of cells characteristic for HLB. Aposymbiotic cells of two *P. putrinum* clones belonging to the two different syngens were experimentally infected by IFs of the new symbiont. The native for the new HLB clone of *P. putrinum* YA111-52 belongs to the syngen 2. Infection of clones from both syngens remained stable for at least two months [20].

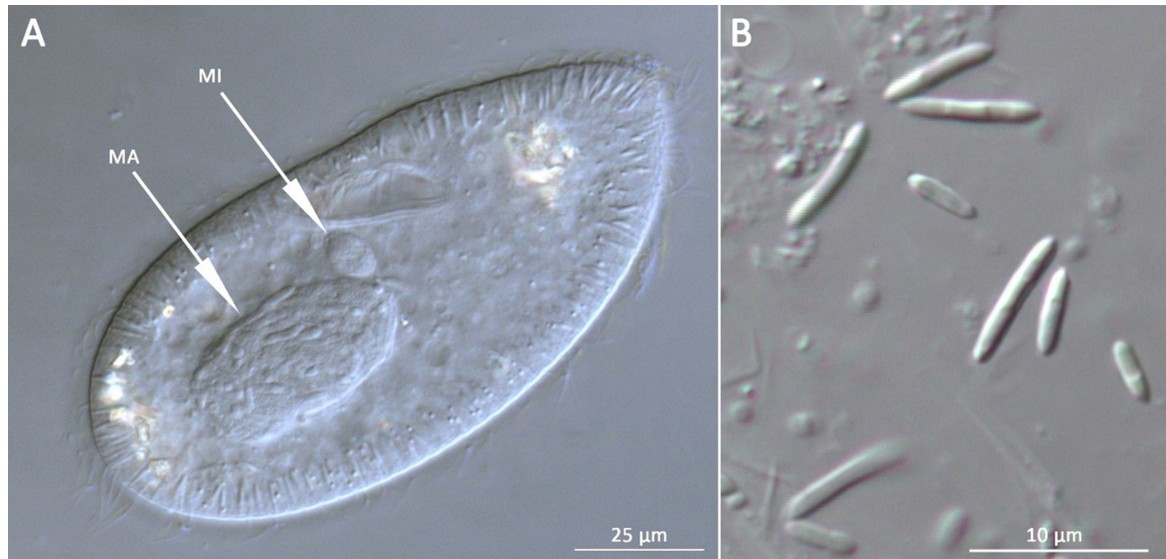

**Figure 1.** (**A**) *P. putrinum* with bacteria in the macronucleus; MA—macronucleus, MI—micronucleus; (**B**) Infectious forms of the new HLB released from the macronucleus.

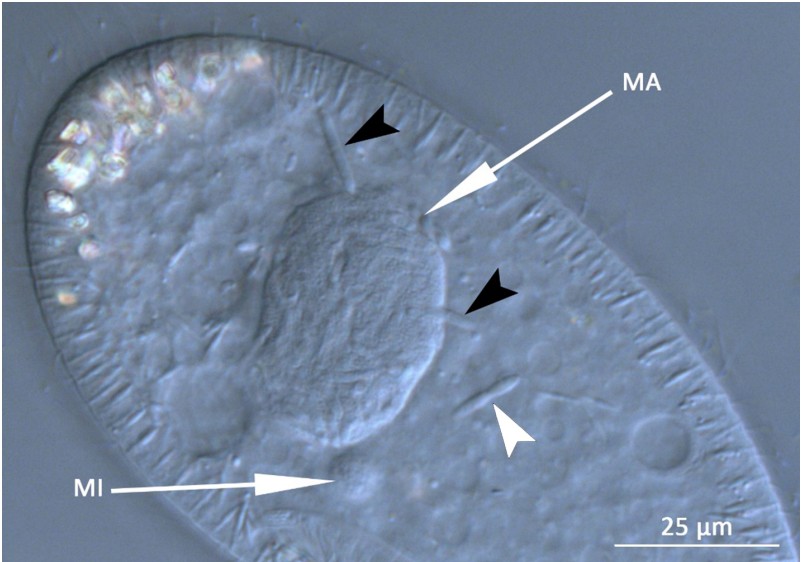

**Figure 2.** *P. putrinum* with bacteria in the macronucleus, individual infectious forms in the cytoplasm shown with black arrowheads, white arrowhead shows IF presumably undergoing a binary fission. MA—macronucleus, MI—micronucleus.

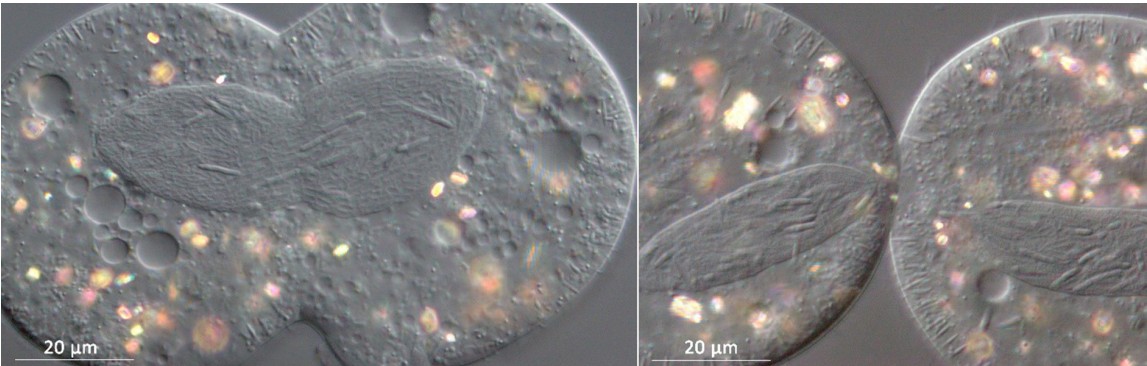

**Figure 3.** *P. putrinum* during the division process. No connecting piece is observed.

### 3.1.1. Molecular Characterization

A total of 26 unique clones were sequenced, among which 21 were assigned to *Holospora*-like bacteria and the remaining 5 were affiliated with *Enterobacteriaceae*, according to the RDP classifier. A 1344–1346 bp long 16S rRNA sequences of the new HLB were deposited at GenBank under the accession numbers MT421875–MT421895. The similarity matrix calculated from multiple sequence alignment shows that the new HLB is closest to "*Ca*. G. shahrazadis" (98–98.2% similarity) and "*Ca*. G. infectiva" (97.4–97.7% similarity).

Using the FISH technique with the sequence-specific probe Gyak567 probe we detected many bacteria in the macronuclei of *P. putrinum* (Figure 4B). The Gyak567 probe bound to bacteria inside the macronucleus in our FISH experiments, thus demonstrating that the characterized 16S rRNA gene sequence had been derived from the new HLB. One of the IFs of the new HLB lies outside the macronucleus in the cytoplasm (marked with the arrowhead), thus confirming our observations, that IFs of the new endosymbiont can escape nucleus. The Eub338 probe was used as a positive control (Figure 4A), it hybridized with the new HLB, as well as with various bacteria in cytoplasm, which most likely are food bacteria.

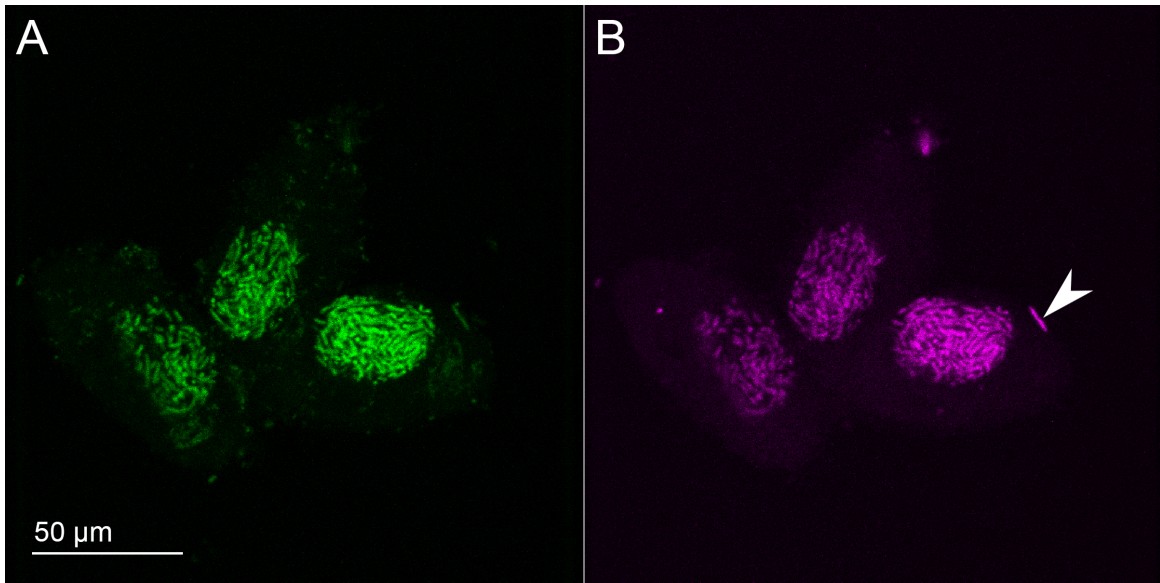

**Figure 4.** Cells of *P. putrinum* with symbionts in macronuclei labeled with the probes Eub338 (**A**) and Gyak567 (**B**). Single IF lying outside the macronucleus is shown with the white arrowhead.

### 3.1.2. Phylogenetic Analysis

The phylogenetic analysis confidently places the new HLB within the "*Ca.* Gortzia" branch as a sister taxon to two other "*Ca.* Gortzia" species, "*Ca.* G. infectiva" and "*Ca.* G. shahrazadis", macronuclear symbionts of *P. jenningsi* and *P. multimicronucleatum*, respectively. However, the level of sequence divergence of 1.8–2.0% and 2.3–2.5% of the new HLB with "*Ca.* G. shahrazadis" and "*Ca.* G. infectiva" respectively suggests that the new HLB is a separate species within the HLB clade and the genus "*Ca.* Gortzia". Two previously described "*Ca.* Gortzia" species show 0.7–0.9% divergence in their published 16S rRNA sequences. The difference of the new HLB with *Holospora* species ranges from 6.9% to 7.2% (Figure 5). Since HLBs are obligate endosymbionts and are not cultivable outside host cells, a complete culture-dependent characterization cannot be provided; hence, we propose the provisional name "*Ca.* Gortzia yakutica".

The phylogenetic tree shows a convincing monophyly of all *Holospora* and "*Ca.* Gortzia" species. (Figure 6).

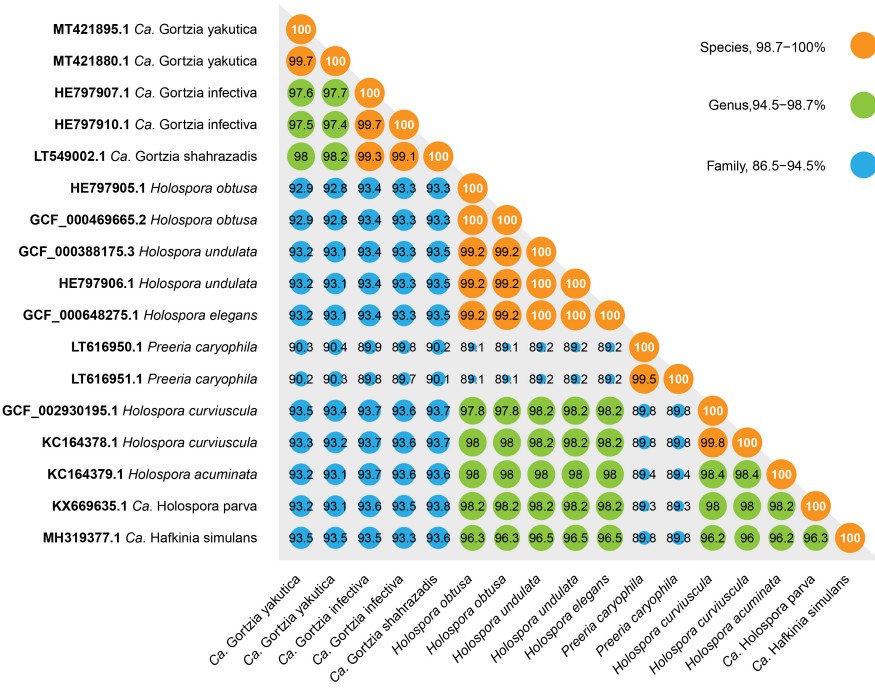

**Figure 5.** Divergence of *Holospora*-like bacteria based on 16S rRNA gene. Pairwise sequence similarity was calculated from the trimmed multiple sequence alignment used for phylogeny inference.

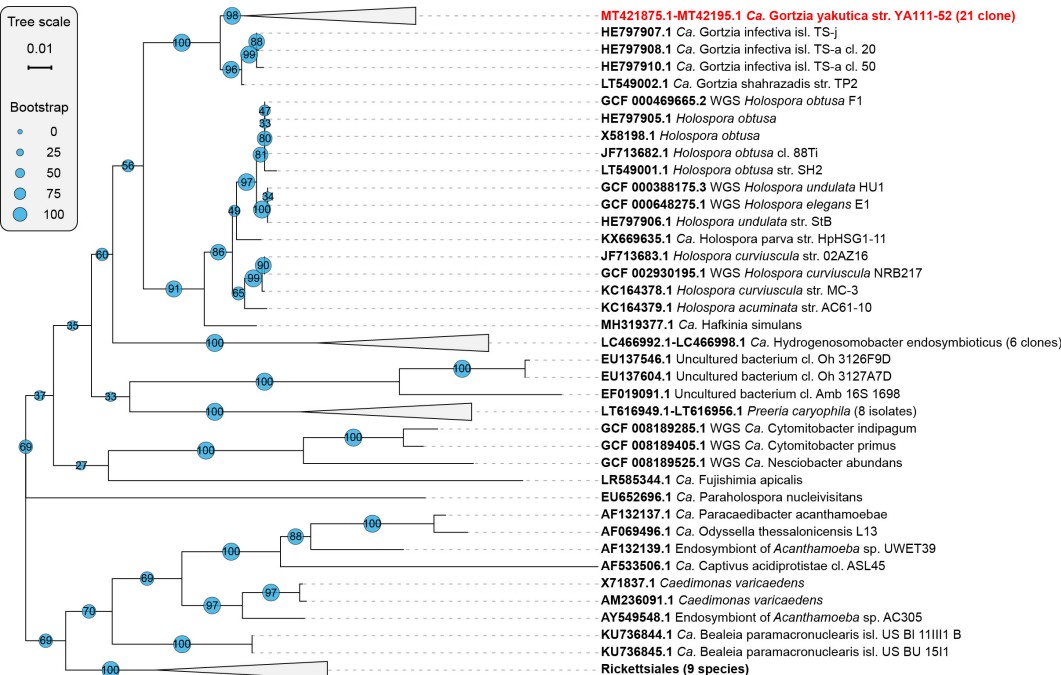

**Figure 6.** Maximum likelihood phylogenetic tree of the order *Holosporales*. Bootstrap support values are shown on each branch. Clones, highly similar isolates, and 9 outgroup sequences from *Rickettsiales* are collapsed. Full list of Genbank sequence identifiers is available in Table S1.

## 4. Discussion

Here, we report a new *Holospora*-like bacterium from the macronucleus of *P. putrinum*. All features of this bacterium, such as morphology, intracellular localization, complex life cycle, host and nuclear specificity, and infectivity, indicate a close relation of this endosymbiont to other HLB. The phylogenetic analysis based on the 16S rRNA gene sequence also shows that the endosymbiont is close to other HLB and belongs to *Holosporaceae* family. Recently described macronuclear symbionts "*Ca.* G. shahrazadis"

and "*Ca.* G. infectiva" are the closest relatives of the new endosymbiont, and together they form a well-supported clade sister to *Holospora* genus. Consequently, we assign this symbiotic bacterium to the genus "*Ca.* Gortzia" and name it "*Ca.* Gortzia yakutica" sp. nov.

As with "*Ca.* G. infectiva" and "*Ca.* G. shahrazadis", the new endosymbiont does not induce formation of the connecting piece during the host division, and can escape into host's cytoplasm; these two features together can be considered to be distinctive for the genus. Another feature to discuss is the manner of transforming IFs to RFs. It is described that IFs of *Holospora* species do not undergo a binary division, but constrict at several points forming a specific chain of cells and divide into several cells simultaneously [6,37]. On the contrary Serra et al. describe binary fission of IFs for the "*Ca.* Gortzia shahrazadis" [12]. We observed a classical IFs chain formation as well as the behavior similar to what is described for "*Ca.* Gortzia" (Figure 2, shown with the white arrowhead). Our knowledge about the transformation of IFs into RFs for the new HLB is limited and based on observations of live *Paramecium* cultures with DIC microscopy, and henceforth is far from comprehensive. At the same time, this phenomenon certainly deserves to be carefully investigated in further studies.

The ability to form the connecting piece also places the new symbiont close to *"H. bacillata"*, *"H. curvata"* and "*H. sp.* from the macronucleus of *P. putrinum*" [1,8]. It is possible that this species was described previously by Fokin et al. as "*Holospora* sp. from macronucleus of *P. putrinum*" from Germany [1,38]. The original description has the information about localization, the shape and sizes of IFs and RFs; the ability to induce the formation of the connecting piece. The new HLB reported here and the endosymbiont from *P. putrinum* reported by Fokin et al. have a similar phenotype (same host and localization, same shape of the cell, both do not form the connecting piece), but the described sizes are different, with the new HLB being notably smaller (e.g., the length of IFs is 12 µm vs. 17 µm). As the culture of "*Holospora* sp. from macronucleus of *P. putrinum*" had been lost precluding a more detailed characterization, it is impossible to establish whether these two endosymbionts belong to the same species.

All HLB have very distinctive morphological and physiological features and form a monophyletic clade within *Holosporaceae* family [10–12,14]. *Holosporaceae* includes four HLB genera—*Holospora*, "*Ca.* Gortzia," "*Ca.* Hafkinia", *Preeria*, and several other genera, which do not share HLB phenotype. Takeshita et al. recently described an endosymbiont from an anaerobic Scuticociliate—"*Ca.* Hydrogenosomobacter endosymbioticus" [39]. This endosymbiont has an uncertain position on the phylogenetic trees based on 16S rRNA genes: according to Takeshita et al. it forms a sister taxon to HLB, but with the low branch support (less than 70%) [39]. In our analysis this species appears within the HLB clade (Figure 6), but the branch support is quite low as well. "*Ca.* H. endosymbioticus" does not have HLB characteristic features discussed above, and its phylogenetic placement would have to be revised when some additional molecular data become available. Another issue arises with "*Ca.* Hafkinia", which was described as a separate genus within HLB, based on the 93.9–94.5% similarity with *Holospora* species [14], whereas our analysis shows 96–96.5% similarity, which places it within the genus *Holospora*.

While phylogenetic analysis based on 16S rRNA gene sequences is undoubtedly useful and widely used to make decisions on bacterial taxonomy, the examples given above show the limitations of such approach. Different thresholds proposed by various authors [15,40], different approaches to multiple alignments and substitution models can affect similarity values and topology of phylogenetic trees. It has been recently demonstrated that complete genome sequences could be used to better define bacterial species [41,42]. Thus, we can conclude that we would be in a much better position to infer the phylogenetic relationships of the HLB clade when complete genomes of *Gortzia* spp. become available.

## 5. Description of "*Candidatus* Gortzia yakutica" sp. nov.

Gortzia yakutica (Gor'tzi.a ya.ku'ti.ca; N.L. fem. n. Gortzia, in honour of Professor emeritus Hans-Dieter Görtz; N.L. fem. adj. yakutica, of or belonging to Republic of Yakutia, the name of the region where the bacterium was first collected).

Obligate macronuclear endosymbionts of the free-living ciliate *P. putrinum*, occasionally can be found in the cytoplasm. Sampled from the freshwater pond in Republic of Yakutia, Russia. Has two life stages: small reproductive forms (1–2 by 2–4 μm) and long infectious forms (1–2 by 7–12 μm, rod-shaped with tapered ends, sometimes slightly curved). No formation of the connecting piece was observed. Basis of assignment: SSU rRNA gene sequence (GenBank accession numbers: MT421875.1–MT421895.1) and positive match with the species-specific FISH oligonucleotide probe Gyak567 (5′-AGGTAGCCACCTACACA-3′).

Type strain is YA111-52 carried by *Paramecium putrinum* YA111-52 (Culture Collection of Ciliates and their Symbionts, CCCS 1024, St. Petersburg State University). Unculturable outside of host cells so far.

## 6. Conclusions

We have reported and characterized a novel species of *Holospora*-like bacteria, "*Ca*. Gortzia yakutica" sp. nov. These intracellular symbionts display several unique biological features such as a complex live cycle and two morphological forms, frequent specificity to the host species and localization inside the host cell, and a distinctive cell structure of infectious forms. Hence they are of interest to evolutionary and infection biology. HLB have been previously reported to be a monophyletic clade within order *Holosporales* based on based on phenotype features and molecular phylogeny. However, we demonstrate phylogenetic placement of genus *Preeria* is uncertain, probably due to the limitations of 16S rRNA-based analysis and the lack of described diversity in the genus currently comprised of only one species. Further research into these fascinating bacteria is well warranted for the understanding of the evolution and systems biology of nuclear endosymbionts.

**Supplementary Materials:** The following are available online at http://www.mdpi.com/1424-2818/12/5/198/s1.

**Author Contributions:** Conceptualisation, A.Y.B., M.S.R., and M.S.G.; data curation, A.Y.B., A.V.P., and S.K.G.; Formal analysis, A.Y.B., A.V.P., and S.K.G.; Funding acquisition, M.S.G.; Investigation, A.Y.B., M.D.L., and S.Z.; Resources, M.S.R.; Supervision, M.S.G. and M.S.R.; Visualization, A.Y.B. and A.V.P.; Writing—original draft, A.Y.B.; writing—review and editing, M.S.R., M.S.G., S.K.G., M.D.L., S.Z., A.V.P., and J.G.; All authors have read and agree to the published version of the manuscript.

**Funding:** This study was supported by the Russian Science Foundation under grant 18-14-00358.

**Acknowledgments:** The study was supported by research facilities of St. Petersburg State University "Center for Culture Collection of Microorganisms", "Molecular and Cell Technologies", and "Chromas".

**Conflicts of Interest:** The authors declare no conflict of interest.

## Abbreviations

The following abbreviations are used in this manuscript:

| | |
|---|---|
| HLB | *Holospora*-like bacteria |
| RF | reproductive form |
| IF | infectious form |
| Ma | macronucleus |
| Mi | micronucleus |

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
