# Peer review of "New Intranuclear Symbiotic Bacteria from Macronucleus of Paramecium putrinum—“Candidatus Gortzia Yakutica”"

_diversity, doi:10.3390/d12050198_

Round 1

Reviewer 1 Report

This paper is a valuable contribution that describes a new endosymbiotic bacteria, Candidatus Gortzia yakutica sp. nov. The conclusions are well supported, and the manuscript is very well written. I enjoyed this study, and I do not have any substantive comments to make. However, I do have a few suggestions which the authors might wish to consider and are in no way meant as criticism:

Minor points:
- L 2-4: I do not think this sentence is needed.
- L9: I suggest using "endosymbiont" instead of "symbiont" whenever possible (i.e., when not redundant in a sentence).
L 10: (and throughout): Please be consistent with the use of SSU rDNA and SSU rRNA (and preferably use SSU rRNA).
L 31: "evidently" may be unnecessary here.

Author Response

This paper is a valuable contribution that describes a new endosymbiotic bacteria, Candidatus Gortzia yakutica sp. nov. The conclusions are well supported, and the manuscript is very well written. I enjoyed this study, and I do not have any substantive comments to make. However, I do have a few suggestions which the authors might wish to consider and are in no way meant as criticism:

We thank the Reviewer 1 for a positive assessment of our manuscript. We have fixed the text according to the minor points listed below:

Minor points:

- L 2-4: I do not think this sentence is needed.

We removed the sentence.

- L9: I suggest using "endosymbiont" instead of "symbiont" whenever possible (i.e., when not redundant in a sentence).

We replaced the word "symbiont" with the more precise "endosymbiont" in all sentences where the localisation is not stated elsewhere. 

L 10: (and throughout): Please be consistent with the use of SSU rDNA and SSU rRNA (and preferably use SSU rRNA).

We replaced "SSU rDNA" with "SSU rRNA" throughout the text.

L 31: "evidently" may be unnecessary here.

Upon other reviewers requests, the introduction was significantly rewritten, and this sentence is not there anymore.

Reviewer 2 Report

The manuscript (ID diversity-781252) entitled “New Intranuclear Symbiotic Bacteria from Macronucleus of Paramecium putrinum” by Beliavskaia and co-authors describes the characterization of a bacterial endosymbiont of Paramecium putrinum. This bacterium lives inside the macronucleus of its hosts and it belongs to a group of bacteria specialized to this habitat. The authors provide light microscopical and FISH analysis as well as 16S rRNA-based phylogenetic analysis. They compare the characteristics of this novel bacterium with those of known endosymbionts with similar life styles and conclude according to the 16S rRNA gee similarities and the position in the phylogenetic tree that the symbiont represents a novel bacterial species in the genus Candidatus Gortzia.

The reported finding of this novel symbiont and the main conclusion – that it represents a novel species belonging to the genus – are highly interesting. The major requirements to report a new Candidatus species are full-filled. My major concerns are that the molecular characterization appears to be sketchy and the reported state of knowledge regarding phylogeny and taxonomy of Holospora and related bacteria is not up to date.

Major comments:

  1. Phylogenetic analysis

1a: In the Discussion and Results section devoted to the Phylogenetic analysis, the line of argumentation why this novel symbiont belongs to the genus Gortzia but represents a new species could be presented more clearly or more strongly. The evidence is all there. They devote one figure/table to sequence comparison but then do not use this information in their argumentation.

1b: There are close members of the genera Gortzia and Holospora absent from the analysis (see comment 7), most prominent Candidatus Hafkinia simulans. As this is placed basal to Holospora (Fokin et al. 2019, PUBMED ID PMID: 30627761), it is interesting how the inclusion of this genus will affect the placement of Gortzia and especially Gortzia yakutia and how similar their 16S rDNA is.

1c: Based on which criteria was the evolutionary model GTRGAMMA chosen?

1d: I am not aware that BLAST algorithms are suitable to determine sequence similarities for species delineation (181). Honestly, I am skeptical regarding the specificity of the outcome as differences in sequences lengths are considered by BLAST and as the authors write, the sequences used of phylogenetic analysis are of different lengths (175). This might impact the similarity values.

  1. Molecular characterization: sequencing

2a: If I understood correctly, the 16S rRNA gene sequence of the symbiont was obtained by PCR with universal bacterial primer (please provide the reference for those) and then cloning. The authors state that they performed an RFLP analysis and sequenced unique clones. I am missing the information how many unique clones were observed, sequenced and what were the results of the sequencing.

2b: Furthermore, I am surprised that a single sequence (62, acc MK209687) was obtained and analyzed (Fig. 5, 6). Is this the consensus of several clones or the sequence of a representative clone? This information should be provided as well as the observed differences between clones and, as this usually happens, which other bacterial sequences have been detected. Furthermore, the sequences of all clones (at least those with the inserted 16S rDNA from the novel symbiont) should be submitted to Genbank to enable others to perform these comparisons.

2c: For completeness, modify the following sentence: “The positive clones were digested with HhaI (Fermentas, Thermo Scientific).” (152) To “DNA obtained via XX (please include method/kit) from positive clones was digested with the restriction enzyme HhaI (Fermentas, Thermo Scientific).”

  1. Molecular characterization: species-specific FISH probe

I am missing how the authors ensured that the new probe Gyak567 is indeed species-specific. The respective method/tool and the obtained result should be included.

  1. Strain: please provide the strain identifier for the here studied P. putrinum YA111-52 also in Material and Method section and the description of the new species. Furthermore, please provide identifiers for the other P. putrinum strains used for crossing (127ff) and include the syngen information (missing from Results)

  1. The amount of cited literature appears very limited, especially in the Introduction section with only 7 citations in total. As the phylogeny and taxonomy of Holospora and related bacteria needs to be updated (see comments below), I strongly encourage to enrich the complete Introduction e.g. with details regarding the life styles of Holospora and related bacteria and to include the appropriate citations.

  1. Phylogenetic update: following symbionts belonging to Holosporaceaea have been recently described (to my information – I encourage the authors to verify this, there might be even more recent descriptions). These symbionts have to be included in Introduction and/or thd Discussion, ideally even in the phylogenetic analysis. If the latter should not be possible, this limitation needs to be specifically addressed in the manuscript. Preeria caryophila
    • Candidatus Holospora parva
    • Candidatus Hafkinia simulans
    • Candidatus Cytomitobacter indipagum
    • Candidatus Cytomitobacter primus
    • Candidatus Bealeia paramacronuclearis
    • Candidatus Hydrogenosomobacter endosymbioticus
    • Candidatus Nesciobacter abundans
    • Candidatus Fujishimia apicalis

  1. Taxonomic update: Holospora caryophila (line 29) has been redescribed as Preeria caryophila (see Potekhin et al. 2018, PMID: 29718229), the genus Caedibacter (line 80) is recognized as member of Gammaproteobacteria (Schrallhammer et al. 2018, PMID: 29426636; Beier et al. 2002, PMID: 12450827). As this manuscript describes a Candidatus new species, the authors should use the correct spelling, i.e. “Candidatus Gortzia yakutica” (7, 107), especially in the formal description. I recommend to correct spelling (with quotation marks) throughout the text, otherwise please indicate that an abbreviated spelling is used.

Minor comments:

  1. Line 2: inhabiting nuclei of Paramecium ciliates and other protists: which other protists aside from Paramecium species harbor HLB?

  1. Line 4: HLB clade is comprised of two genera: this info is outdated (see comment 7)

  1. Line 58: IFs reach macronucleus and begin to divide: in literature this process is usually described as differentiation or similar while reproduction (binary division) is considered to be limited to the reproductive form of HLB. Is this a true difference in the behavior of G. yakutia to other HLB or should this sentence be rephrased?

  1. Line 85: There is an error with the citation Fokin [1], does not fit to the bibliography (here [1] Fujishima 2009)

  1. Line 97: The authors remark on the possible variability between 16S rDNA operons, which is a valid point in case of organisms possessing multiple operons. For completeness sake they might consider mentioning the number of 16S operons in the sequenced Holospora genomes and, possibly, their variability. Alternatively, they should mention that for intracellular bacteria the copy number is often low, usually 1.

  1. Line 100: Later publications refer to a threshold of 98.7%, e.g. Yarza et al. 2014 PMID: 25118885, a value which has been used since.

  1. Line 116: Unculturable outside of host cells. Add “ so far”.

  1. Check that Genus names are given (not only abbreviated) when mentioned the first time, but might be abbreviated later on in the text. That is at present not uniform, e.g. P. multimicronucleatum (33, 71) but no Paramecium multimicronucleatum. Please fix the organisms names in line 51.

  1. While I like the graphical depiction of sequence similarity by spot size (Fig 5), I think the symbols alone lack the informational content needed from this figure. I recommend to include the numerical values in the spots. If this should not be possible, please provide either only the values e.g. as supplementary figure or as replacement of the current fig. 5.

  1. Candidatus should not abbreviated as ca. but with capital C (Fig. 5).

  1. The accession number should be indicated (Fig. 5), ideally the sequences of the type strains should be compared.

  1. Update organism names in Fig. 6 according to comment 7.

Reviewer 3 Report

The manuscript by Beliavskaia and coauthors presents the characterisation and taxonomic description of a novel candidate species of endonuclear symbiont of the ciliate Paramecium putrinum. These results offer some improvement in the knowledge in the diversity and evolution of Holospora-like bacteria, thus being of potential interest for specialists, and possibly also for a more general public interested in symbiosis and Holosporales. Overall, the experimental data provided seem to be supportive of the main conclusions, however, in my opinion the data analysis and validation, and the overall writing of the manuscript (in terms of presentation with respect to the literature, experimental details provision, interpretation and discussion of results) require significant improvements before it can be published.

Major concerns:

-The introduction is in my opinion too succinct, incomplete in terms of references, and for some features out of date. While the phylogenetic and taxonomic diversity of HLB bacteria is of course central for the manuscript, I think HLB features from a more general perspective should be either added or more underlined, in particular the high peculiarity of their life cycle and features with respect to other bacteria, even closely related, and what is known concerning the interaction and effect on the host. Potentially also references to other symbionts of Paramecium, in particular Holosporales, could be considered. Large passages of the text (e.g. lines 15-23) are unreferenced (e.g. for the papers reporting molecular data of Holospora) and/or contain incomplete/incorrect/out-of-date information (here a list of things I noticed, might not be exhaustive). If I remember correctly, although the full 16S rRNA gene of H. elegans was probably not amplified and sequenced, its sequence is available from the genome, I think this piece of information should be included in line 17 adjusting the sentence accordingly. Holospora caryophila, according to novel molecular data, has been redescribed as member of a novel genus of HLB (Preeria caryophila), which is particularly relevant considering that it is more distantly related to Holospora than “Ca. Gortzia” itself (even in the authors’ phylogeny in Fig.6). Also, another genus of HLB, namely “Ca. Hafkinia”, was recently described, and is more closely related to Holospora than “Ca. Gortzia”, making incorrect the sentence of line 31 that “Gortzia is evidently a sister taxon of Holospora” (which, even if it was supported by up-to-date results, would in any case be not so appropriate phrasing in my opinion). Also, another candidate Holospora species was recently describe (“Ca. Holospora parva”), requiring an update of Holospora species number and listing in the text.

-The authors state that they have designed a species-specific probe that was used for diagnosis and description of the novel species. However, no specificity test (e.g. RDP, TestProbe on Silva database) was reported, this should be fixed, as it obviously precludes the intepretation of FISH data. I also think that the specific mismatches/gaps with previously sequences “Ca. Gortzia” species should be reported for reference of the specificity even with respect to close relatives.

-Phylogeny and identity calculation procedures are not entirely convincing for me. The authors reported that they used an automated software (mafft) for alignment, and then filtered the positions. To my own experience, 16S rRNA genes contain many insertions/deletions which hamper proper automatic alignment (especially of relatively distant and fast-evolving organisms such as Holosporales symbionts) of all position. The best option in those cases is to refine extensively the alignment manually taking into account predicted secondary structure of the rRNA. This can result in significant improvements in the accuracy of the analysis (even more so with respect to blast concerning the identity, which is not intended for such phylogenetic/taxonomic application). Also, how did the authors choose GTR+G model? This should be done following the estimate of appropriate software (e.g. jmodeltest, modeltest-ng). I am confused by the species selection and report, as the authors refer to the figure for the accession, but the outgroup sequences were not reported, which is important to evaluate the approach used. I would also suggest an additional phylogeny on Bayesian Inference approach for comparison, this would strengthen the results obtained. Concerning the identity, the values should best calculated on a curated identity matrix used for phylogeny, instead of on blast hits, because these are pairwise and not directly comparable among themselves, which is instead critical to compare with taxonomic thresholds (see below)

-I don’t think the Discussion section is properly organised and written, in my opinion it should be extensively revised, here I mention some major features. The comparison with a Holospora sp. found in P. putrinum previously is very interesting and should be expanded according to all available data. Clearly, in the absence of molecular data for that one it is impossible to get to a final conclusion, but it would be valuable to know whether the data are compatible or not with a synonymy. The presence of infectious forms in the cytoplasm is very interesting and should be discussed. It is misleading (and in case not appropriate for Result section) to define this feature as “uncharacteristical” for HLB, as something similar was observed in the closely related “Ca. Gortzia sharazadis”. The two conditions should be critically compared. Also, in figure 2 the pointed cytoplasmic cell at the bottom seems to be to bear some central constriction, as if it was possibly undergoing division, what do the authors think? Last, while it is not entirely clear to me what the authors mean by lines 94-105, for what I can understand I am quite doubtful on the appropriateness of their statements. The life cycle of HLB is clearly a distinctive feature of this monophyletic clade, but at the same time internal genus-level subvisions are strongly supported by molecular genetic divergence and phylogeny. Concerning sequence divergence in 16S and thresholds, I advise the authors using the most commonly accepted references for taxonomic purposes of species and above taxonomic ranks, e.g. (Kim et al. Int. J. Syst. Evol. Microbiol. 64:346–351; Yarza et al. Nat. Rev. Microbiol. 12:635–645). Also their novel data should be discussed accordingly (according to the presented data, I agree that this is a novel “Ca. Gortzia” species), rather than referring to 97%, which, to my knowledge, is mostly used for OTU findings on metagenomic studies (frequently on partial genes). Furthermore, mentioning variations among different copies of 16S rRNA genes seems not appropriate for HLB and obligate symbionts in general, as these, due to genome streamlining, have one (as in all published Holospora) or typically at most two copies.

-The taxonomic rules for writing names do not appear to have been followed throughout the text. Some species of Holospora (“H. bacillata”, “H. curviuscula” and others) were not validly described, and should be reported in italics, but within quotation marks. Candidate names should be always reported within quotation marks, e.g. “Candidatus Gortzia yakutica”, with only Candidatus word in italics. I would discourage referring to these candidate taxa in the text without the candidate notation as on the contrary the authors sometimes did (e.g. Gortzia yakutica, or Gortzia); this can be acceptable, but should be done consistently throughout the text, ideally explaining in the first occurrence that for brevity it will be used this notation instead of the complete name. In any case, the taxonomic description at the end should report the full Candidatus notation.

-Names of organisms, including the HLB Preeria caryophila, are not updated in the tree, other examples include Caedibacter caryophila (=Caedimonas varicaedens), Trojanella thessalonicensis (= “Ca. Odyssella thessalonicensis”). Also, Holosporaceae and maybe other families could be better evidenced.

Additional comments:

-Line 39: should be “16S rRNA gene” and “in situ” written in italics

-Line 44: I think at this point of the text a more neutrally descriptive expression than “inhabit” would be convenient, such as “were seen in”

-Line 55: be careful on the typing of quotation marks, also I think the correct abbreviation is Holospora sp. This sentence is in any case more appropriate for discussion than results

-Line 63: I think it would be convenient to report from the beginning the best hit(s) and identity on blast, distinguished from more accurate identity calculations based on aligned sequences (see major comments above)

-Line 66: I think the authors should also address the point whether the EUB338 probe fluorescence evidenced other intracellular bacteria distinct from the symbiont of interest, and whether these are inside food vacuole possibly. I see some faint fluorescence in fig4a, which should be mentioned by the authors.

-Figure 5: I could not understand the legend. In any case, I suggest reporting actual numbers instead of colours only, being more informative

-line 85: I think the original publication by Fokin should be definitely referenced here, instead of the cited review, which I am noted even sure accounted for the data of interest

-line 122: remove “the” before lettuce

-line 123: put “live” instead of “living”

-line 124: what do the authors mean by “research park”?

-line 129: maybe better “an equal”

-line 130-136: this is results and/or discussion. Where does this data come from? In general I think all other culture tested should be mentioned and clarified whether they were molecularly characterized. In the absence of an effective characterisation of the analyses strain (advisable), at least the compatibility with confirmed strains would be indirect evidence

-line 140: “Four times” in which timings?

-line 154: were sequencing primers the same as amplification primers?

-line 159: please write Bacteria in place of Eubacteria

-line 172-173: This sentence concerning the maximum accepted length of the sequence (1500bp), roughly corresponding to the typical full-length of the 16S rRNA gene, is confusing. From my experience with Holosporales, I assume the authors intended here to get rid of the relatively frequent sequences harbouring insertions (presumed, and in case of Caedimonas, verified, to be excised from the mature rRNA), which obviously increase the total length, and might disturb the alignment, especially in a non-manually revised approach. This might be anyway cryptic to most readers, and should be clarified.

Round 2

Reviewer 2 Report

The authors provide a revised version of their manuscript and a detailed point-by-point response addressing all mentioned issues.

I have only 4 small remarks.

- Omit citations in the abstract

- Citation Preeria (Preer and Preer 1982) incorrect, replace by Potekhin et al. 2018 (line2)

- Citation [5] Gong et al. 2014 refers to a very interesting endosymbiont of Paramecium, but it is not a member of Alphaproteobacteria (line14)

- Figure 5 is cited prior to figure 4 -> reconsider order of figures

Author Response

  • Omit citations in the abstract

Citation removed from the abstract

  • Citation Preeria (Preer and Preer 1982) incorrect, replace by Potekhin et al. 2018 (line2)

Fixed citations

  • Citation [5] Gong et al. 2014 refers to a very interesting endosymbiont of Paramecium, but it is not a member of Alphaproteobacteria (line14)

Removed

  • Figure 5 is cited prior to figure 4 -> reconsider order of figures

I moved the Methods section to the beginning of the article following the editor's request, and the first figures to be mentioned in the text are Figure 5 and Figure 6 (mentioned in the Methods section). I don't think there's a need to change the order of pictures, because they follow the logic of the narrative of the Results section. 

Reviewer 3 Report

The revised manuscript by Beliavskaia and coauthors provided significant improvements in the analyses and the presentation of the results and literature, addressing the major points raised.

I have still some minor points and comments regarding writing that I believe should be addressed before publication:

-line 2: say better “of Paramecium and other ciliates”, as “Ca. Hafkinia” is in Frontonia

-line 3: the second Candidatus word should be in italics as well

-line 4: I disagree that HLB species have “strict” specificity for host species (e.g. the case of Preeria), I would suggest rephrasing as “some degree of specificity”

-line 7: please substitute “known” with “described”, as the Holospora sp. from P. putrinum was already known though not formally described

-line 14: I would suggest rephrasing, because ref 5 is not related to Holosporaceae

-line 42-44: I could not understand to which sequences specifically the sentence is referring to

-line 54: I would suggest writing “a freshwater pond”

-line 57: maybe it is better “there might be an intermediate state”

-Fig 2: please state in the legend the difference between black and white arrowheads

-line 69: if I understood correctly, presented data are novel, in such case, no reference is needed here

-line 105: it should be ‘sister to Holospora and “Ca. Hafkinia” genera’, as shown in figure 6

-line 109-110: I suggest rephrasing the second part of the sentence, considering that if I well remember cytoplasmic localisation is a feature of “Ca. Gortzia sharhazadis”, but not of “Ca. Gortzia infectiva”

-line 119: it should be written “Holospora sp.”

-line 135-141: in my opinion this sentence should be rephrased a bit. The two cases present as similar are rather different. For “Ca. Hydrogenosomobacter”, there seems to be an issue of correct phylogenetic placement with incongruence between different phylogenetic analyses, possibly due to long-branch attraction artifacts between fast-evolving sequence. On the other side, the sister phylogenetic relationship between genera Holospora and “Ca. Hafkinia” seems robust according to available data, while the point would be on whether the identity is within or below the threshold for distinguishing two genera in a bordeline case, probably strongly affected by site selection masks in different studies.

-lines 142-149: I don’t agree completely with this sentence, and suggest some adjustments. While obviously 16S rRNA gene sequences display obviously limitations, and, due to the larger amount of data, genome sequences can ideally used to overcome some of those, I would discourage saying that this “settles” the issue, considering that many of the issues presented, chiefly different approaches for alignments leading to different results and the choice of different thresholds, clearly persist, though possibly in a different form. Also, I would generalise the last part, saying genome of “Ca. Gortzia” and other HLB/Holosporaceae.

-line 150: the name should be reported in the complete as “Candidatus Gortzia yakutica” in the description

Author Response

We would like to thank the Reviewer for a thorough analysis of our manuscript.

We have changed the text according to minor comments 1-13 and 16. 

-line 135-141: in my opinion this sentence should be rephrased a bit. The two cases present as similar are rather different. For “Ca. Hydrogenosomobacter”, there seems to be an issue of correct phylogenetic placement with incongruence between different phylogenetic analyses, possibly due to long-branch attraction artifacts between fast-evolving sequence. On the other side, the sister phylogenetic relationship between genera Holospora and “Ca. Hafkinia” seems robust according to available data, while the point would be on whether the identity is within or below the threshold for distinguishing two genera in a bordeline case, probably strongly affected by site selection masks in different studies.

We agree, that these two examples are not similar, although they serve as a demonstration of the limitations of the 16S based phylogeny, that's why we have placed them together. We have replaced "similar" with "another".

-lines 142-149: I don’t agree completely with this sentence, and suggest some adjustments. While obviously 16S rRNA gene sequences display obviously limitations, and, due to the larger amount of data, genome sequences can ideally used to overcome some of those, I would discourage saying that this “settles” the issue, considering that many of the issues presented, chiefly different approaches for alignments leading to different results and the choice of different thresholds, clearly persist, though possibly in a different form. Also, I would generalise the last part, saying genome of “Ca. Gortzia” and other HLB/Holosporaceae.

We agree with the Reviewer that technical issues certainly can and do introduce bias into the analysis of whole-genome sequences. At the same time, we believe that whole-genome sequences contain much more evolutionary signal and thus represent the best option currently available to reconstruct phylogenetics of bacteria. We have removed the part of the sentence where we state that it would settle the problem.